# BODIPYs in PDT: A Journey through the Most Interesting Molecules Produced in the Last 10 Years

**DOI:** 10.3390/ijms231710198

**Published:** 2022-09-05

**Authors:** Miryam Chiara Malacarne, Marzia Bruna Gariboldi, Enrico Caruso

**Affiliations:** Department of Biotechnology and Life Sciences (DBSV), University of Insubria, Via J.H. Dunant 3, 21100 Varese, Italy

**Keywords:** 4,4-difluoro-4-bora-3a,4a-diaza-s-indacene, BODIPY, photodynamic therapy, photosensitizer

## Abstract

Over the past 30 years, photodynamic therapy (PDT) has shown great development. In the clinical setting the few approved molecules belong almost exclusively to the porphyrin family; but in the scientific field, in recent years many researchers have been interested in other families of photosensitizers, among which BODIPY has shown particular interest. BODIPY is the acronym for 4,4-difluoro-4-bora-3a, 4a-diaza-s-indacene, and is a family of molecules well-known for their properties in the field of imaging. In order for these molecules to be used in PDT, a structural modification is necessary which involves the introduction of heavy atoms, such as bromine and iodine, in the beta positions of the pyrrole ring; this change favors the intersystem crossing, and increases the ^1^O_2_ yield. This mini review focused on a series of structural changes made to BODIPYs to further increase ^1^O_2_ production and bioavailability by improving cell targeting or photoactivity efficiency.

## 1. Introduction

### 1.1. Photodynamic Therapy (PDT)

Photodynamic therapy (PDT) [1,2] is an innovative technique that combines three components which are individually harmless but which when combined cause damage to nearby biomolecules (Figure 1).

Nowadays, PDT is used for diverse pathologies, and it is gaining interest in cancer treatment [2,3,4,5,6,7,8]. PDT involves three components: a photosensitizer (PS), a molecule that has a high degree of unsaturation that is able to capture the energy provided by the second component, light at an appropriate wavelength; this energy is then transferred to the third component, molecular oxygen [1,9,10]. Specifically, once irradiated, the PS in the excited electronic state is highly unstable and will therefore tend to return to the ground state. The return to the ground state can occur through the emission of fluorescence, an excellent property in the clinical setting for imaging or photodetection, and/or heat [11]. Alternatively, and based on physico-chemical characteristics, the PS can undergo an intersystem crossing (ISC); with this step the PS will go into a more stable triplet state (^3^PS*) with an inverted spin of an electron. ^3^PS* can return to the ground state by emitting phosphorescence or it can interact with molecular oxygen (O_2_) [12] thus leading to the formation of singlet oxygen (^1^O_2_) in what is called a Type II reaction [13,14,15]. Type I reactions can also occur; in this case the PS interacts directly with a substrate [16,17,18,19,20], such as the cell membrane or a molecule, thus transferring a proton or an electron to the substrate to form radicals which will then react with O_2_ to form the superoxide anion (O_2_^•−^), the hydroxyl radical (OH^•^) and hydrogen peroxide (H_2_O_2_) [6,21,22,23,24]. Type I and Type II reactions can occur simultaneously, even though photoreaction of type II is predominant [13,25,26]. ROS that are generated by type I and II reactions are mainly responsible for PDT-induced cell death [27,28]. These species are extremely toxic and, once formed, can strongly reduce malignant cells through necrosis, apoptosis, autophagy [4,10,29,30,31], and, as reported by many authors, can cause inflammatory/immune responses [32,33,34,35,36,37,38].

One of the three crucial elements of PDT is the PS [39]. PSs used are generally colored substances that share, as a common feature, an extensive electronic conjugation by a high number of conjugated double bonds to allow the interaction of π electrons with low energy radiations such as visible light.

The ideal PS for therapeutic application should have a selective accumulation in the tumor tissue [11]. The accumulation is favored by a certain degree of lipophilicity which favors cellular uptake but at the same time is also favored by a certain degree of hydrophilicity which facilitates its administration; the two concepts can be exemplified with the term amphiphilia [7,11,39]. PS must also have no or negligible intrinsic toxicity to preserve healthy tissues [6,39]. From a chemical point of view, PS should be easily synthesized in pure formulation, have a high molar extinction coefficient [7] and a high quantum yield of ^1^O_2_ [40]. The ideal PS should be activated at wavelengths of between 680 and 800 nm for a deeper penetration of the light [41].

Over the years various generations of PSs have been obtained to meet all the requirements of the ideal PS. Historically, the PSs are divided into first, second and third generation [1,11,42].

The first generation includes naturally derived PSs belonging to the porphyrin family that were developed in the 1970s and early 1980s. The first preparations for use in PDT were based on a fairly complex mixture of Hematoporphyrin-derived (HpD) porphyrins [1,43]. HpDs showed better tissue selectivity for tumors and less photosensitizing potential of the skin [44,45], but they have limitations such as low chemical purity, poor tissue penetration and skin hypersensitivity to light [1,39,46,47,48].

In the 1980s, to overcome those limitations, several hundreds of substances with potential for photosensitizing had been proposed, however, only a few were used in clinical trials. The second generation includes synthetic PSs such as 5-aminolevulinic acid, benzoporphyrin derivatives, texaphyrins, chlorins, bacteriochlorins, BODIPYs, anthraquinones, xanthenes, cyanines and curcuminoids [1,5,39,49,50,51,52,53]. Those compounds show higher chemical purity, a higher yield of ^1^O_2_ formation and better penetration [6]. The strong limitation turns out to be water solubility.

Using second generation PSs as a basis and binding them with cargos to allow targeted accumulation in the tumor site, the third generation of PSs was obtained [10,54]. In this category PSs are conjugated with organic and inorganic polymers, nanoparticles, liposomes, monoclonal antibodies, antibody fragments, protein/peptides (such as transferrin, EGF, somatostatin and insulin), carbohydrates, folic acid, and many others [1,24,55,56]. These modifications lead to enhance targeting and absorption of the PS in tumoral and drained-vasculature cells within the tumors [57]. Many PSs belonging to this generation are currently under development or undergoing clinical trial.

### 1.2. BODIPYs

The first member of 4,4-difluoro-4-bora-3a,4a-diaza-s-indacene dyes (hereafter abbreviated to BODIPYs) was reported by Treibs and Kreuzer in 1968 [58] although relatively little attention was given to their discovery until the end of the 1980s [59]. The potentials of this dyes for biological labeling and in many other science fields was later recognized and a series of study brought attention on them.

In addition to the high fluorescence emission (Φ), the BODIPYs are characterized by a high molar extinction coefficient (ε) in the visible region, limited changes by environmental conditions (e.g., polarity of the solvent or pH) [60], by a high lipophilicity [61] and a good resistance to photobleaching (the phenomenon of chemical destruction of a fluorophore by light [62]).

The absorption and emission characteristics of the BODIPY can be adjusted by adding suitable substituents on the main structure of the BODIPY itself. The effect is particularly evident when the substituent is bonded to one of the carbon atoms of a pyrrole ring [63]. On the contrary, the presence of an aromatic group in position 8 (meso) exerts a very weak effect, mainly explained by the weak electronic interaction between the aromatic ring and the main structure of the BODIPY [55].

Any structural difference on the *meso*-aryl group does not greatly affect the intensity and wavelength of the absorption and fluorescence bands, however the degree of polarity of this part can modulate the degree of amphiphilicity of the molecule.

The presence of substituents in positions 2 and 6 of pyrrole (*β* positions) involves some evident effects; for example, the presence of alkyl groups or heavy atoms has direct repercussions on the energy of the HOMO and LUMO orbitals and on the triplet state so as to influence the absorption and emission phenomena of UV-Vis radiation [64,65].

In addition to the well-known fluorescence that makes these molecules excellent biomarkers, a new important application field of BODIPYs concerns photodynamic therapy (PDT) as they can be used as photosensitizers (PS), after appropriate modifications [61].

A fundamental modification to increase the probability of ISC and the quantum yield of ^1^O_2_ formation is the introduction in some specific positions of a heavy atom, such as iodine or bromine. A heavy atom can quench the fluorescence and facilitate the transition to triplet state. Heavy atoms either in solution (external Heavy Atom Quenching HAQ) or incorporated in the molecule (internal HAQ) assumed to quench fluorescence by perturbating S1 state via spin orbit interactions that modifies electron’s atomic energy levels. The transition between the T_1_ state and ground state are forbidden due to the spin changes, this results in a relatively long-lived and reactive excited configuration that can interact with other molecule forming complexes or radicals [40,56,61,65].

Iodine seems to severely quench fluorescence, suggesting that the intersystem crossing efficiency (Φ_ISC_) has been enhanced by the internal heavy atom effects. Interestingly 2I-BODIPYs showed a narrow peak in NIR spectrum that is characteristic of ^1^O_2_ generating molecules [40,66,67,68,69,70,71]. The incorporation of more heavy atoms in BODIPY core does not effectively enhance the ^1^O_2_ quantum yield; on the contrary it appears to increase dark toxicity.

## 2. BODIPYs Studies

Substituents are generally attached in positions 4 (boron center); 3,5 (α-positions); 1,2,6,7 (*β*-positions), and 8 (*meso* position). In this context, we will show new structural derivatives of BODIPY core and their strategy to improve delivery and efficiency [56,59,66]. Principal modifications are the addition of groups that modify chemical and physical properties (hydrophilicity, absorption wavelength, ISC efficiency, etc.), or improve the interaction with biological medium (environment-response, association with targeting molecules, etc.) [56,59,66,72].

Modifications to their structures enable the tuning of their fluorescence characteristics, enhancing their capacity for ^1^O_2_ generation. The stability of the core structure and its tendency to maintain photochemical properties even with complex groups as substituents in some positions makes BODIPYs an interesting class of compounds for PDT [56,59,72].

### 2.1. Structure-Activity

One of the important aspects in the study of the various PSs is to evaluate the relationship that exists between the chemical structure and the photodynamic activity of the PS; in fact, this could allow the design of targeted syntheses. In recent years many authors have tried to analyze this type of relationship in various series of PS such as porphyrins [73,74], the pyropheophorbides [75] and phthalocyanines [76].

Even in the case of BODIPYs this type of approach is extremely important for their application in PDT [40,71,77,78,79,80,81,82,83,84,85].

Recently, we characterized a panel of twenty-four new BODIPYs all united by an aromatic ring in *meso* position and by the presence of iodine atoms in *β*-positions [86]. The panel shows different atoms or groups as substituents of the aromatic ring. All compounds were characterized from a physico-chemical point of view (generation of ^1^O_2_, fluorescence and lipophilicity) and for their activity on the human ovarian carcinoma cells SKOV3. A quantitative structure-activity relationship (QSAR) analysis was also performed to understand how the substituent could affect PS activity.

Data obtained demonstrate that the presence of an aromatic ring is essential to obtain a high production of ^1^O_2_ and high phototoxicity of the compounds. Of the twenty-four compounds analyzed, two showed high activity (compounds **2** and **8**).

The first is characterized by the presence of a methoxy group in the *para* position while the other has two chlorine atoms in the *ortho* and *ortho’* position (Figure 2). The differences in regioisomerism and in the electronic effect confirm the absence of correlations between the substituents and the efficacy of the photo-induced action.

The presence of iodine atoms in the *β*-positions of the pyrrole determine a high production of ^1^O_2_. The IC_50_ were in the range of 1-2 nM for the BODIPY derivatives with the exception of compounds **2** and **8** (Table 1).

Based on these results, the effect of chains of different length on the hydroxyl groups was evaluated. Two BODIPYs with the presence of hydro-carbon chains of different lengths placed in position 4 (*para*) of the phenyl ring placed in *meso* position of the core were analyzed [87]. Specifically, compounds **3** and **4** (Figure 3) have carbon chains consisting of four and eight atoms, respectively, to obtain a different degree of lipophilicity of the two molecules. This difference should make it possible to establish a correlation between the length of the alkyl chain and the photodynamic activity. A bromine atom was presented at the end of each chain to make these molecules more versatile and eventually allow the bond with nanoparticles [88].

The analyzes carried out confirmed that the presence of alkyl chains of different lengths affects the degree of lipophilicity and consequently also the photodynamic effect. Compound **3** has a fair degree of lipophilicity associated with an effective photodynamic action on tumor cells; on the contrary, the more lipophilic compound **4** has a lower photodynamic activity probably due to the formation of aggregates in aqueous medium (Table 2).

As already highlighted the main problem with these PSs leads to their solubility in aqueous medium. Belfield et al. designed a BODIPY decorated in meso position with a polyethylene glycol chain to enhance solubility and to prevent aggregation in aqueous solution [89].

Compound **3** (Figure 4) was characterized as regards absorption, fluorescence and ^1^O_2_ production. The ability to produce ^1^O_2_ was evaluated by direct measurement of near infrared luminescence. The quantum yield of ^1^O_2_ generation is higher than that of the standard used as a control (0.93 for **3**, 0.82 for acridine used as control). IC_50_ were estimated at 10 µM against Lewis lung carcinoma cells pre-treated with compound **3** and exposed to a light dose of 3.5 mW/cm^2^ (Table 3).

In addition, the cell death mechanisms induced following photodynamic treatment were evaluated and necrosis prevails over apoptosis process. The results obtained show how it is feasible to obtain a BODIPY with a higher degree of hydrophilicity with at the same time excellent characteristics for use in PDT.

### 2.2. Singlet Oxygen Generation

Critical factor for the activity of BODIPY is certainly the production of ^1^O_2_. The generation of ^1^O_2_ in solution requires a photosensitizer (PS) which is converted to the triplet excited state (intersystem crossing, ISC) upon irradiation. The triplet state PS transfers energy to molecular oxygen in a type II process to produce ^1^O_2_. The heavy atom effect has been a useful chemical approach to improve ISC in several molecules including BODIPY chromophores.

Lincoln et al., and Durantini et al., investigated the different photostability and PS efficiency for PDT of two BODIPYs bearing an acetoxymethyl substituent in the *meso* position and bromine or iodine atoms in positions 2 and 6 of the scaffold [90,91].

The presence of the substituent acetoxymethyl in meso position improves the photostability of the PS. (Figure 5). It was also observed that the presence of iodine (**I_2_BOAc**) or bromine (**Br_2_BOAc**) atoms in both *β*-positions affects the production level of ^1^O_2_. The PS bearing the two bromine atoms produced a lower level of ^1^O_2_ than that bearing the iodine atoms. The production of ^1^O_2_ by the bromine compound conferred a slight improvement in stability mainly attributable to the lower amount of oxidant generated (Table 4).

Cytotoxicity studies conducted with the HeLa tumor cell line have shown that the compound bearing the bromine atoms is the most active. The localization of these compounds is mainly in the lipid membranes of the cell. Tests to evaluate the photodynamic activity of bacterial inactivation were carried out against Gram-negative *Escherichia coli*. The results obtained showed that both molecules are able to photo-kill *E. coli* already from a concentration of 5 μM, suggesting that both molecules have a PDT potential against *E. coli* and other Gram-negative strains.

A similar comparison between the two types of halogens bound in *β*-positions of the BODIPY was also carried out by Epelde-Elezcano et al., who studied the photo-physical properties of a series of BODIPYs bearing either two bromine atoms or two iodine in *β*-positions of the pyrrole of the BODIPY skeleton. Specifically, the measurement of ^1^O_2_ production was carried out with an indirect assay using 9,10-dimethylanthracene (DMA) as a chemical probe and with a direct determination of the luminescence at 1276 nm of the ^1^O_2_ with a NIR detector. The work demonstrates how the iodine atom favors the ISC over the bromine atom. In the same work the authors evaluate the impact of the *meso* substituent, observing how an electron donor substituent tends to decrease the production of ^1^O_2_ compared to an analogous derivative bearing an electron withdrawing substituent. Furthermore, a strong decrease is observed in the case of electron donor substituents when there is free rotation of the substituent which favors the internal conversion process to the detriment of the ISC and therefore of the production of ^1^O_2_ [92].

Turan et al., focused on synthesis of BODIPY derivatives that could be applied to enhanced fractional photodynamic therapy [93]. In this therapeutic strategy and to prevent photo-induced hypoxia, the light is administered intermittently (fractional PDT) to allow the replenishment of cellular oxygen with an increase in the time necessary for the effective therapy. The study reported the synthesis of a BODIPY containing 2-pyridone (Figure 6) which in the dark conditions, through a thermal cycloreversion, becomes a source of ^1^O_2_ [94,95,96,97,98].

When excited at a wavelength of 650 nm, molecule **6** (**Pyr**) generates ^1^O_2_ which is partially stored in the form of 2-pyridone-endoperoxide (**7** or **EPO**). When irradiation is stopped **EPO** will undergo thermal cycloreversion to produce ^1^O_2_ in the absence of light (Table 5) [99,100].

The molecules were tested against human cervical HeLa cells. Due to the low solubility, the compounds were prepared as micellar structures using the non-ionic surfactant cremophor EL [101]. IC_50_ was estimated at 8.6 nM and 49 nM for compound **7** and **6**, respectively. Results demonstrates how the continuous release of ^1^O_2_ by the photosensitizer during the light-dark cycles has to be considered as a positive feature if we compare the cytotoxic effects with those of conventional photosensitizing agents.

Zou et al., instead focused on how the effect and configuration of the substituents of heavy atoms are important for ^1^O_2_ generation [102]. A series of BODIPY derivatives was synthesized with one or two BODIPY units connected via a benzene ring; PSs thus obtained were then halogenated in *β*-positions with bromine or iodine atoms. The ability to produce ^1^O_2_ and cytotoxic efficacy in HeLa cells was evaluated for each of the compounds.

The best producers of ^1^O_2_ are compounds **4** and **6**, consisting of one and two units of iodinated BODIPY, respectively. Moreover, the insertion of two or more heavy atoms does not significantly affect the improvement of the ^1^O_2_ produced (Table 6).

The synthesized compounds were then included in NPs to improve water solubility, required for cytotoxicity tests against HeLa cells. The MTT test showed that the most active compound is **6** (Figure 7). Given the good results obtained in vitro, compound **6** was also used for in vivo tests. In vivo, treatment with compound **6** led to a dramatic decrease in tumor growth and no tumor recurrence was observed. Biodistribution analysis showed that compound **6** accumulated in tumor cells and the liver. The selective irradiation of the tumor zone alone ensures that the liver is spared from possible photodynamic damage.

Pang et al., considered the dimerization of two BODIPYs to increase ^1^O_2_ generation (Figure 8). In this case, the second BODIPY molecule is covalently linked in the beta position of the first BODIPY and can assume two positions, either orthogonal as in compounds **2b** and **2d** or angular (angle of about 30–40°) in compounds **2a** and **2c** [103].

Following the synthesis, the authors took into consideration some spectroscopic properties and the ^1^O_2_ production evaluated both through direct and indirect determination (Table 7).

The highest efficiency was observed for orthogonal meso–b linked dimer which confirm the data that Cakmak et al., had previously reported as well [101].

### 2.3. Uptake and Targeting

As reported by many authors [104,105,106,107,108,109,110,111], the aspect of cellular uptake and targeting has always been a very important aspect for all families of photosensitizers. Trying to increase cellular uptake as well as to make cellular penetration selective between healthy and tumor tissues is of primary importance, consequently cellular uptake and targeting are fundamental and interrelated factors.

#### 2.3.1. Uptake

Shivran et al., synthesized three water soluble glucose conjugated BODIPYs and subsequently tested them on human lung cancer A549 cells. Of the three compounds, compound **4** features a glycosylated styryl appendix in position C-3 (Figure 9) [68]. The choice to insert a glucose on the PS is linked to the greater ability of the cancer cells to accumulate sugars.

The emission spectrum of this compound is typical of a BODIPY with a high molar extinction coefficient and a high fluorescence emission. The introduction of the styryl group determines a bathochromic shift both in the absorption and in the maximum emission with respect to the BODIPY in which this group is absent. The IC_50_ value of all the compounds was evaluated and compound **4** is endowed with the greatest cytotoxicity (Table 8).

Moreover, compound **4** accumulated rapidly in tumor cells. In addition, cell exposure to compound **4** led to apoptosis which is the prevailing cell death mechanism. Overall, the work in question demonstrated how it is possible to obtain a compound capable of having a selective capacity towards cancer cells in an economical and fast way.

Kuong Mai et al. [112] synthesized a water soluble BODIPY starting with a halogenated alkyl azide BODIPY to which a lactose motif is linked through an easy and straightforward CuAAC Click reaction [113]. The coupling of the lactose motif causes the compound to be soluble in water (Figure 10).

The emission and absorption spectra are consistent with those of the BODIPY family and the production of ^1^O_2_ is high. The photodynamic activity of the PS was evaluated on three different tumor cell lines: human hepatoma Huh7, cervical cancer HeLa and breast cancer MCF7 and IC_50_ ranging from 0.6 to 0.5 µM (Table 9).

However, cell uptake was low, whatever the cell line tested, probably due to the presence of iodine atoms on the core of the BODIPY.

#### 2.3.2. Targeting

Active targeted therapy of cancer refers to targeting the surface molecules on cancer cells using a small molecule to deliver cargo for therapeutic or diagnostic purposes. The cargo can be a cytotoxic drug, an imaging probe, or a photosensitizer [114].

##### Lysosomal Targeting

In 2019 Wang et al., obtained a BODIPY able to specifically targeting lysosomes [115]. The molecule obtained (**BDPI-Lyso**) has iodine atoms to generate ^1^O_2_ and morpholine for targeting specifically the lysosomes (Figure 11) [116,117,118].

Physico-chemical analyzes showed that the absorption and fluorescence of the compound are slightly greater than that of the common BODIPYs due to the modification of the sulfur heteroatom. Moreover, under irradiation, the derivative produced higher level of ^1^O_2_ in acidic versus physiological pH (Table 10).

This leads to the assumption that BODIPY exhibits greater photo-toxic effects in acid compartments such as lysosomes than in neutral organelles. In fact, the BODIPY derivative was localized in lysosomes as demonstrated by cell imaging performed with hepatoma Bel-7402 cells. Moreover, IC_50_ was estimated at 0.4 µM after cell pretreatment with the molecule and light exposition for 30 min duration.

Similarly, Li et al. [119] synthetized a BODIPY (**MBDP**) (Figure 12) containing morpholine for lysosomal targeting and iodine atoms on the BODIPY core to produce ^1^O_2_. Triethylene glycol monomethyl ether benzaldehyde was added through the condensation of Knoevenagel to obtain the desired compound [120].

The analysis of the absorption spectrum of the compound showed that there is an intense cui bands at 660 nm. At 660 nm, the compound under LED light produces high levels of ^1^O_2_ (Table 11).

Cytotoxicity tests were performed on the MCF7 cell line with an estimated IC_50_ at 0.2 µM. The colocalization experiments confirmed the effective lysosomal localization of the compound. It was also possible to determine how, following irradiation, the lysosomes containing photosensitizer break down following ^1^O_2_ and ROS generation. In addition, the BODIPY derivative could also be used for NIR-PDT. 

##### Reticulum Targeting

Another target is the plasma reticulum. The BODIPY **10** (Figure 13) bearing an analogous glibenclamide motif obtained by Zhou et al., has precisely the goal of targeting towards the plasma reticulum [121].

The absorption spectrum of the compound exhibits the band at 669 nm. The quantity of ^1^O_2_ produced by the compound is very low when results were compared to those obtained with the reference molecule, zinc(II) phthalocyanine. The photodynamic activity has been studied on HeLa and HepG2 cells. The light source used is a 300 W halogen lamp on which a colored glass filter has been placed plus a cut-on at λ = 610 nm. The IC_50_ values obtained in both cell lines are lower than 0.2 μM (0.09 μM in HeLa and 0.16 μM in HepG2). Such low concentrations are linked to the high uptake of the compound by the cell lines being analyzed (Table 12).

The compound was found to be associated with the endoplasmic reticulum using confocal microscopy when the results were compared to those obtained with cells treated with an analog molecule without glibenclamide. The analog molecule did not accumulate in reticulum.

##### Mitochondrial Targeting

In 2021 Bhattacheryya et al., thought of binding a iodine-BODIPY to a metallic ternary system to increase its solubility in water, an essential factor for in vivo applications of highly lipophilic molecules [122,123] (Figure 14).

Curcumin was further coordinated within the system. Curcumin is well known for its tumor specific activity [124,125,126] but also as a photosensitizer. Curcumin in the native form has poor bioavailability and is susceptible to hydrolysis under cellular pH conditions thus reducing its therapeutic efficacy. However, curcumin bound to transition metal ions became stable and the metal complexes show significant mitochondrial localization [127]. In this work the authors show that the complex **3** having green light harvesting di-iodinated BODIPY and curcumin for mitochondrial targeting have an interesting photodynamic activity in human breast cancer (MCF7) via disruption of mitochondrial membrane (Table 13).

##### Cell Membrane Targeting

The targeting can be implemented not only towards cytoplasmic organelles but also towards receptors present on cell plasmic membrane.

The asialoglycoprotein receptor (ASGP) is expressed exclusively in the liver and was considered by Li et al., to design a BODIPY that specifically targeted it [111]. For this purpose, the author has synthesized a new macro-molecular PS equipped with hydrophilic appendix capable of selectively targeting liver cancer cells. The compound obtained is a galactose-functionalized BODIPY-based macromolecular photosensitizer **p(GEMA-co-BODIPYMA)-2I** with good solubility in aqueous environment (Figure 15).

The compound proved to be a good ^1^O_2_ producer. Biological tests were carried out on HepG2 and NIH3T3 cell lines and first concerned the specific binding towards liver cells and subsequently the cytotoxic activity on them was evaluated (Table 14).

The compound was observed in the cytoplasm of HepG2 cells as demonstrated by confocal microscopy using fluorescence emission of iodine atoms. Previous studies carried out by the same group [128] have shown that the removal of galactose does not allow targeting towards HepG2 cells. The results therefore suggest that the internalization of **p(GEMA-co-BODIPYMA)-2I** depends mainly on the ASGP receptor, over expressed in HepG2 cells. The photodynamic efficacy of the compound was then evaluated. Cell survival was less than 20% following treatment with a concentration lower than 20 µM. In addition, cell exposure to the BODIPY derivative triggered cell to apoptosis. In contrast, the compound did not accumulate in NIH3T3 lacking ASGP receptor expression and the cells did not undergo cell death after light exposure.

Tyrosinase is a key regulatory enzyme in the biosynthesis of melanin through melanogenesis and is localized in the membrane of melanosome. Tyrosinase level is closely correlated with the malignancy level, so it can be considered as a biomarker of melanoma cell. Phenylthiourea (PTU) is one of the most important and well-known tyrosinase inhibitors. Kim et al., designed a BODIPY derivative with PTU pendant to enhance the PDT efficacy against melanoma cell line (Figure 16).

Cellular uptake studies demonstrated the specificity of the binding of the compound to tyrosinase overexpressed in murine B16F10 cells [129] (Table 15).

Epidermal growth factor (EGFR) is another target since it is expressed in diverse tumor cells. In recent years, a series of sequences able to bind specifically to this receptor have been identified [130]; Zhao et al., selected the sequence namely D4 which was bound to polyethylene glycol and subsequently placed on a BODIPY (Figure 17) [131].

Conjugation with the peptide (**8**) shows a slight shift of the emission and absorption bands with respect to the precursor without the peptide. The quantum yield of fluorescence is five times lower than that of the compound without the peptide (Table 16).

Photodynamic activity data on HepG2 human carcinoma cell, known to express EGFR [132,133] studies show that the conjugate is more active than the precursor without the peptide this is probably due to the presence of a positive charge on the peptide sequence which should favor a faster and more efficient cell internalization [134]. Uptake kinetic demonstrated a fast absorption of the molecule over 24 h, suggesting that the presence of triethylene glycol and the positive charge of the peptide sequence increase the internalization of the compound.

Burgess et al. worked on targeting the receptor tyrosine kinase [135,136,137]. Starting from a series of previous experiments in which specific conjugates for the receptor tyrosine kinase with cytotoxicity were identified, Burgess decided to hook these conjugates to a BODIPY to obtain a molecule (**IY-IY-PDT**) (Figure 18) usable in PDT [136].

Following the satisfactory results of the physico-chemical characterization, the molecule obtained was tested on NIHT3T wild-type (NIHT3T-WT) and over cell lines expressing the receptor tyrosine kinase (NIHT3T-TrkC). It was possible to detect a significant photo-induced cytotoxicity effect in over-expressing cells following treatment with the compound **IY-IY-PDT** (IC_50_ = 0.35 μM) (Table 17).

In cellular localization it states that the compound under analysis is mainly located at the level of the lysosomes while it does not accumulate at the mitochondrial or endoplasmic reticulum level by exploiting the tyrosine kinase receptor.

The following year, the same author tested the same compound on breast cancer cell lines expressing receptor tyrosine kinase [136]. The data obtained also in this case affirm that the compound is more assimilated by the cells that express the receptor and also in this case the localization of the compound is at the lysosomal level. The first in vivo tests on mice are also carried out in this article which showed that there is a high decrease in tumor size in mice starting at four days of illumination.

In the next article published in 2016, the author reports an increased antitumor immune response after treatment with the conjugate and illumination of the area of interest [137]. The overall analysis of the data reported in the three articles published by the author leads to the hypothesis that the compound **IY-IY-PDT** acts as a therapeutic agent capable of stimulating the immune system, which makes it an excellent candidate for use in clinical practice of cancer treatment.

### 2.4. Sensitive Activity

Another area in which the research was concentrated was that of obtaining BODIPYs capable of being activated in specific conditions such as acid pH, typical of cancer cells, or in the presence of particular molecules such as glutathione (GSH).

Radunz et al. evaluated the activity of a pH-activable fluorescent ^1^O_2_ generating BODIPY dyes (**3b**), which can be used as PS and for bioimaging (Figure 19) [138].

To ensure that the compound was able to produce a high level of ^1^O_2_, the core of the BODIPY has iodine atoms in *β*-positions. The responsive part of the pH is instead represented by a phenolic substituent placed in the meso position of the core.

From the titrations carried out in the pH range between 4.5 and 8.2 it is observed that the absorption is minimally influenced by the pH while the intensity of the fluorescence decreases drastically at basic pH. The production of ^1^O_2_ is shown in Table 18.

As a result of the irradiation the compound showed an activity with an IC_50_ on the HeLa tumor cell line equal to 70 nM. The authors repeated the photodynamic experiment by adjusting the intracellular pH to a value of 5.5 or 7.5 before illumination [138]. The results obtained showed that cell survival was lower for cells with an acidic environment compared to those in a neutral environment (cell survival was 10% vs. 70% at a PS concentration of 0.1 μM).

Cao et al. reported the synthesis of a GSH-responsive BODIPY (**8a**) consisting of three modules: a BODIPY-based photosensitizing chromophore, a BODIPY-based quencher and a bio-reducible disulfide linker (Figure 20) [139].

The compound thus obtained has shown to have excellent photophysical and photochemical characteristics such as a strong absorption in the visible region and in the NIR as well as high photostability. Overall, the compound does not show fluorescence, but after the addition of 10 μM of GSH which breaks the disulfide bond, the fluorescence is recovered.

In vitro studies have been conducted on three cell lines (A549, H22 and HeLa) known to have high intracellular concentrations of GSH. The compound is rapidly internalized by cells and effectively activated by intracellular biothiols. In the presence of light, the activity expressed as IC_50_ is of the order of µM. The compound is mainly localized in mitochondria where it is believed that GSH-mediated cleavage may also occur (Table 19).

In vivo experiments on H22 tumor-bearing mice showed high selectivity in tumor tissue and efficacy in inhibiting tumor growth.

Jiang et al., synthetized a BODIPY responsive to both pH and thiol groups [140]. Specifically, compound (**7**) has two ferrocenyl moieties attached to a combination of ketal- and disulfide-linkers (Figure 21). The former is cleaved in acid conditions, while the latter with cellular GSH.

Fluorescence and ^1^O_2_ production of the molecule depends on the two ferrocenyl groups, in fact at acid pH and in the presence of GSH, which break the ketal and the disulfide bond, an increase in fluorescence and ^1^O_2_ is observed (Table 20).

Photodynamic activity studies on the MCF7 cell line showed that compound **7** has dithiothreitol (DTT) dependent behavior (IC_50_ halves when DTT goes from 2 μM to 4 mM).

In vivo experiments were conducted using nude mice carrying HT29 human colorectal carcinoma. From the fluorescence images it is observed that compound **7** is localized only and exclusively in the area of the tumor, suggesting that the compound is activated only and exclusively at the tumor level where there is the cleavage of the two linkers, by means of pH and DTT, and consequent detachment of the two ferrocenyl motifs.

## Figures and Tables

**Figure 1 ijms-23-10198-f001:**
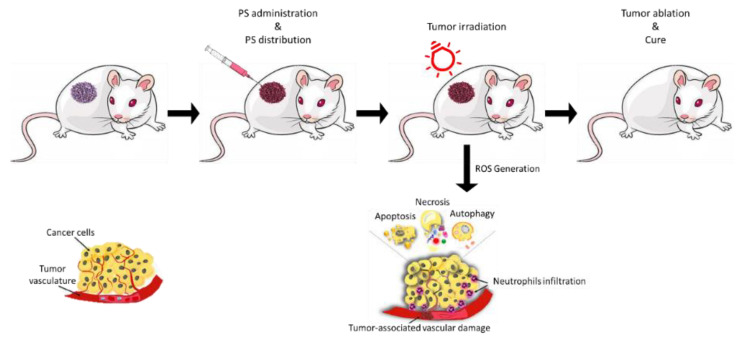
Schematic PDT Treatment: photodynamic strategy involves photosensitizer (PS) administration (local or systemic injection). Light irradiation is applied depending on the drug-light interval (time necessary to drug accumulation within the tumor). The activation of PS leads to the generation of ^1^O_2_ and ROS that cause cancer cell death through apoptosis, necrosis or autophagy.

**Figure 2 ijms-23-10198-f002:**
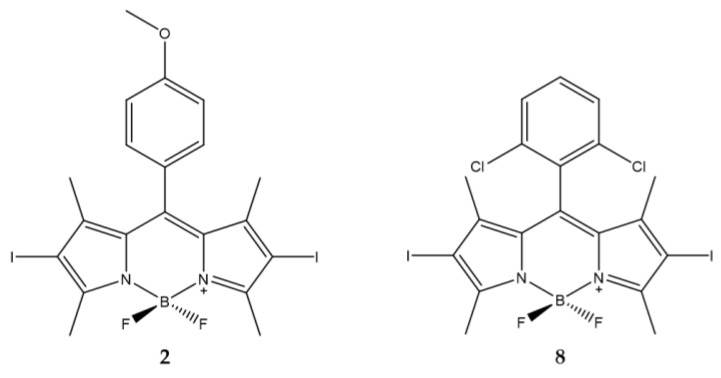
Chemical structures of compounds **2** and **8** [86].

**Figure 3 ijms-23-10198-f003:**
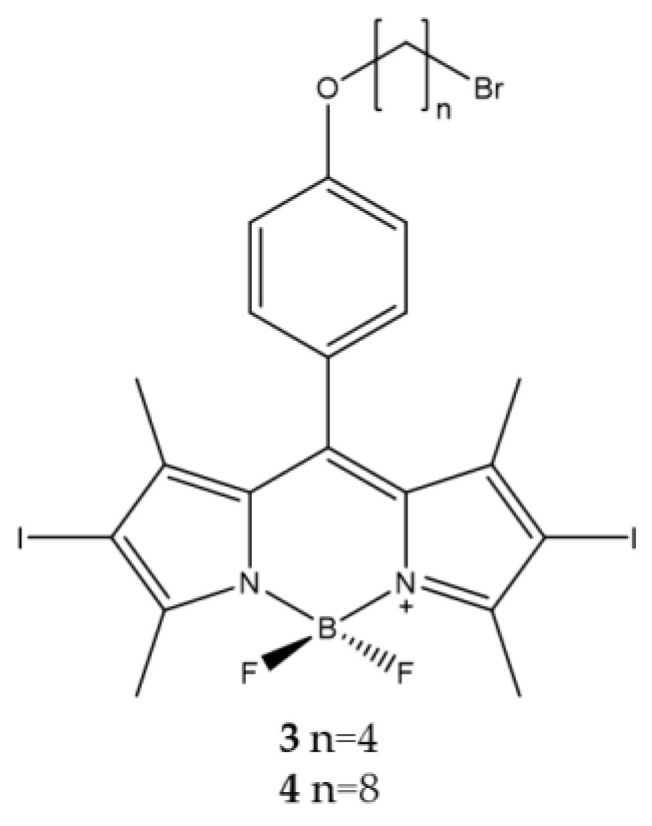
Chemical structures of compounds **3** and **4** [87].

**Figure 4 ijms-23-10198-f004:**
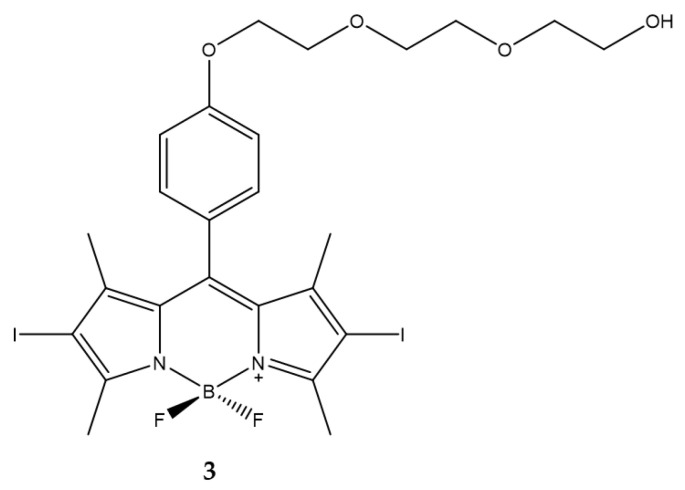
Chemical structures of compound **3** [89].

**Figure 5 ijms-23-10198-f005:**
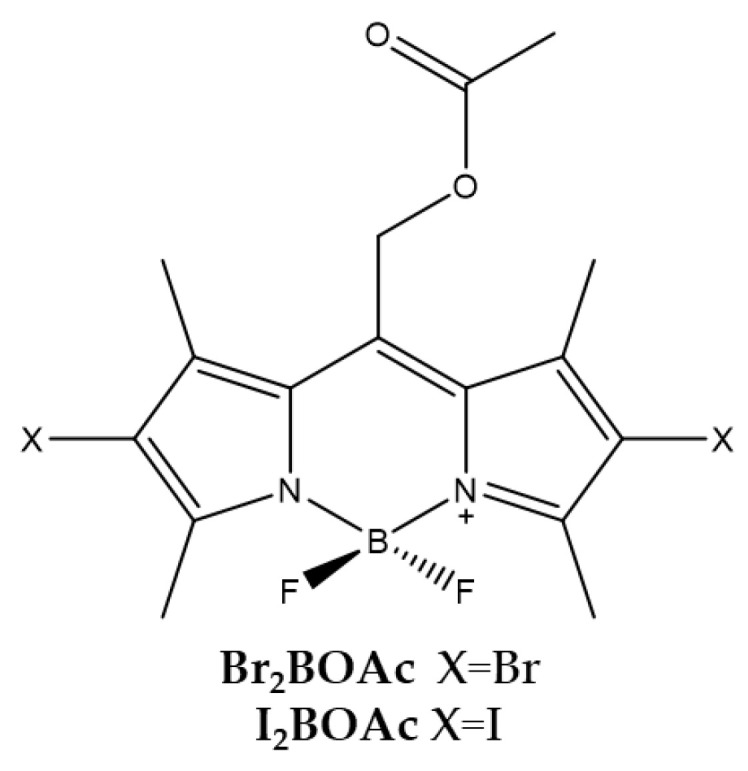
Chemical structures of compounds **Br_2_BOAc** and **I_2_BOAc** [90,91].

**Figure 6 ijms-23-10198-f006:**
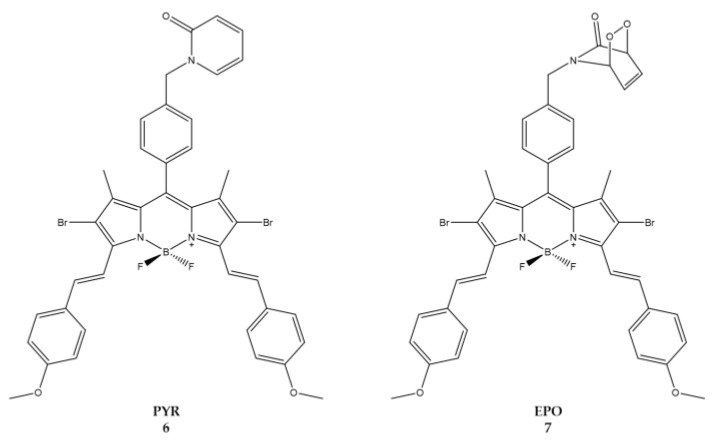
Chemical structures of compounds **6** (**PYR**) and **7** (**EPO**) [93].

**Figure 7 ijms-23-10198-f007:**
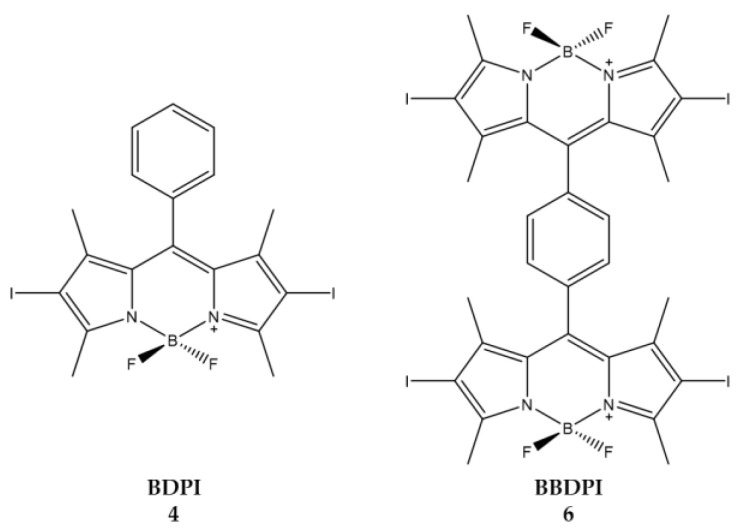
Chemical structures of compounds **4** (**BDPI**) and **6** (**BBDPI**) [102].

**Figure 8 ijms-23-10198-f008:**
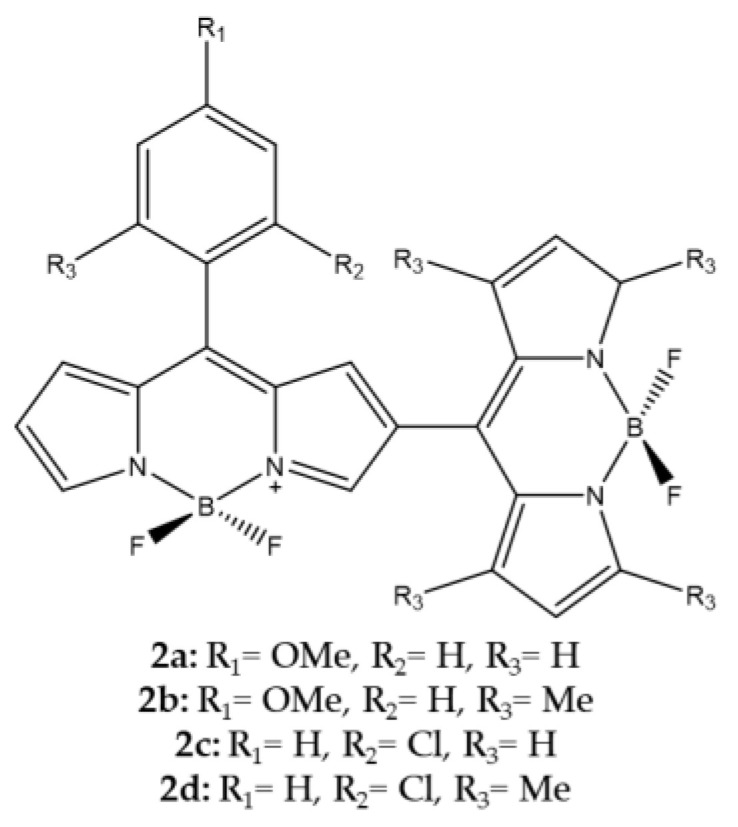
Chemical structures of compounds **2a**–**d** [103].

**Figure 9 ijms-23-10198-f009:**
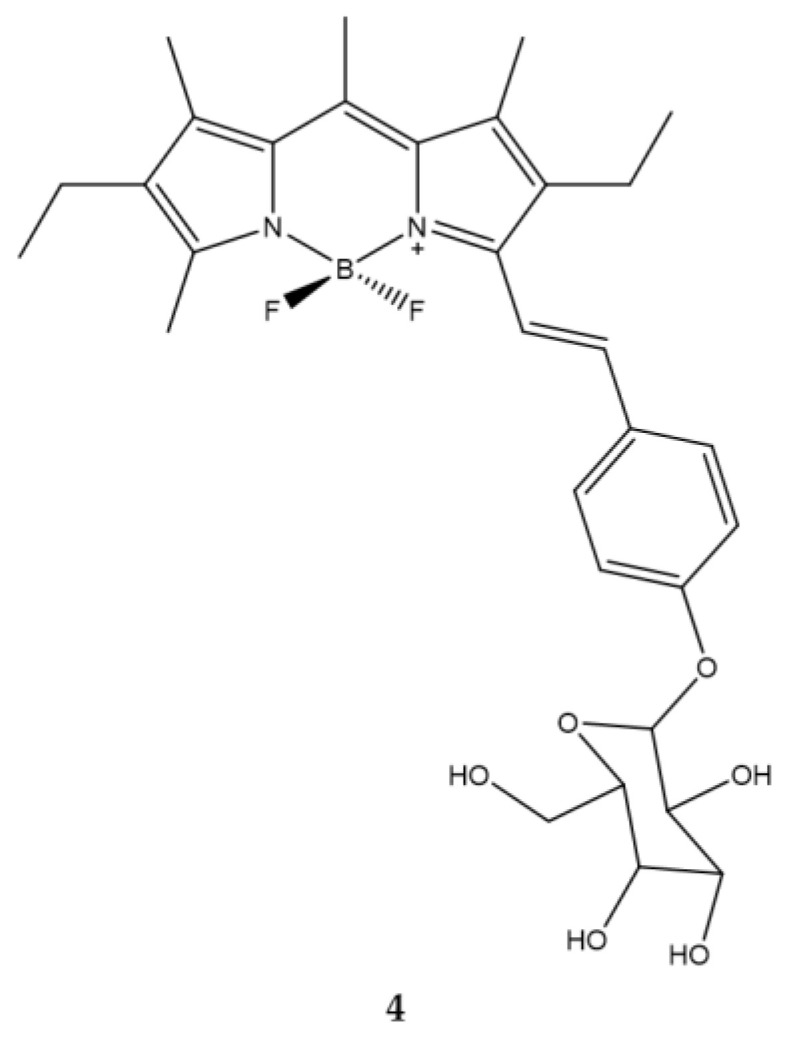
Chemical structures of compound **4** [68].

**Figure 10 ijms-23-10198-f010:**
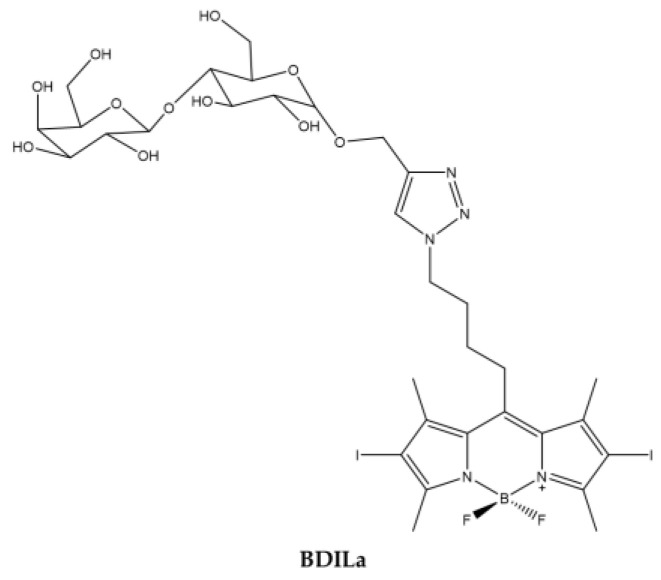
Chemical structures of compound **BDILa** [112].

**Figure 11 ijms-23-10198-f011:**
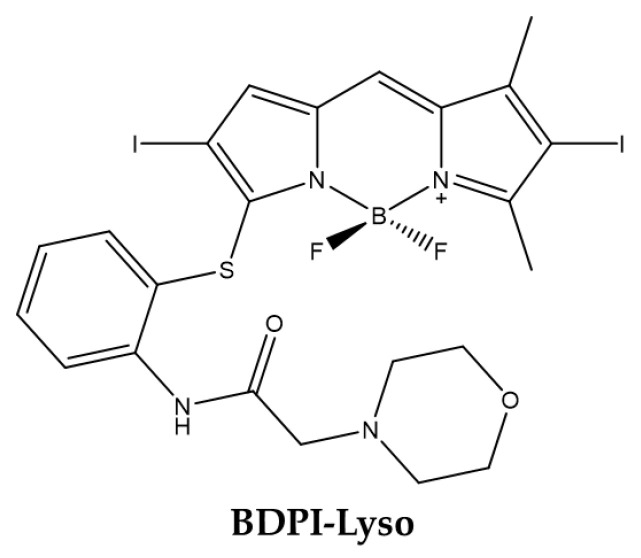
Chemical structures of compound **BDPI-Lyso** [115].

**Figure 12 ijms-23-10198-f012:**
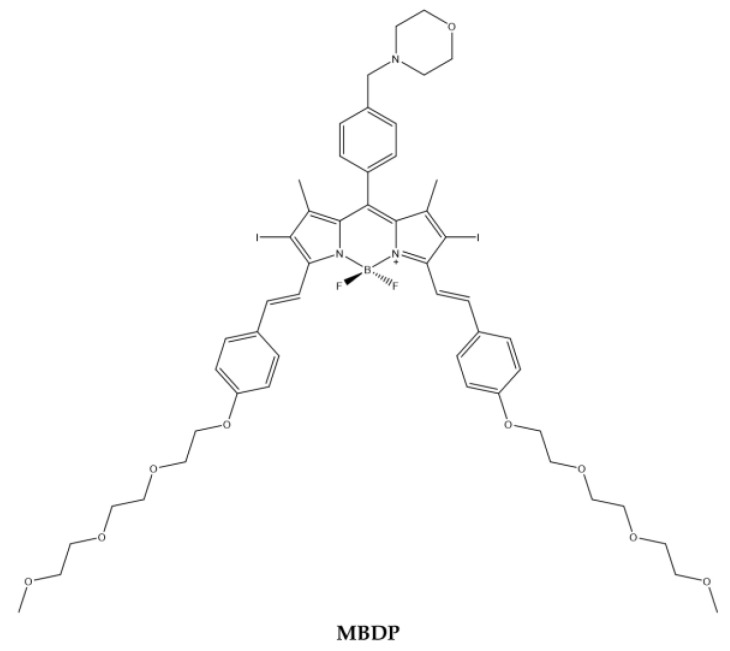
Chemical structures of compound **MBDP** [119].

**Figure 13 ijms-23-10198-f013:**
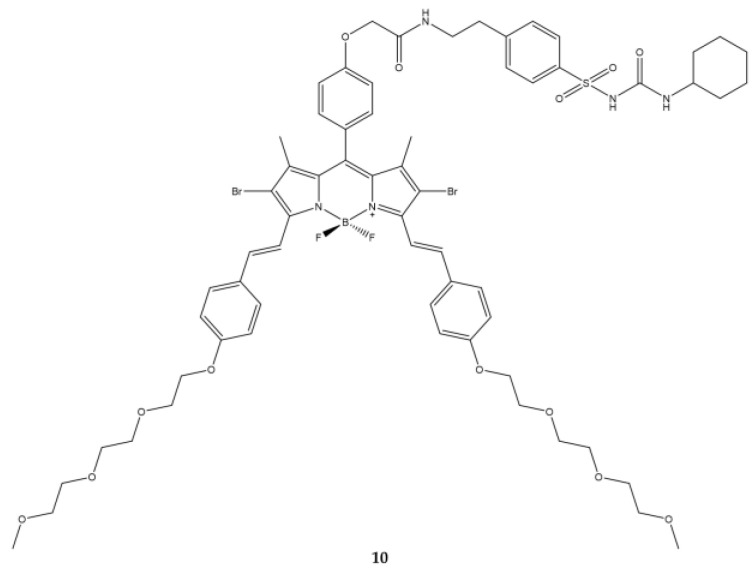
Chemical structures of compound **10** [121].

**Figure 14 ijms-23-10198-f014:**
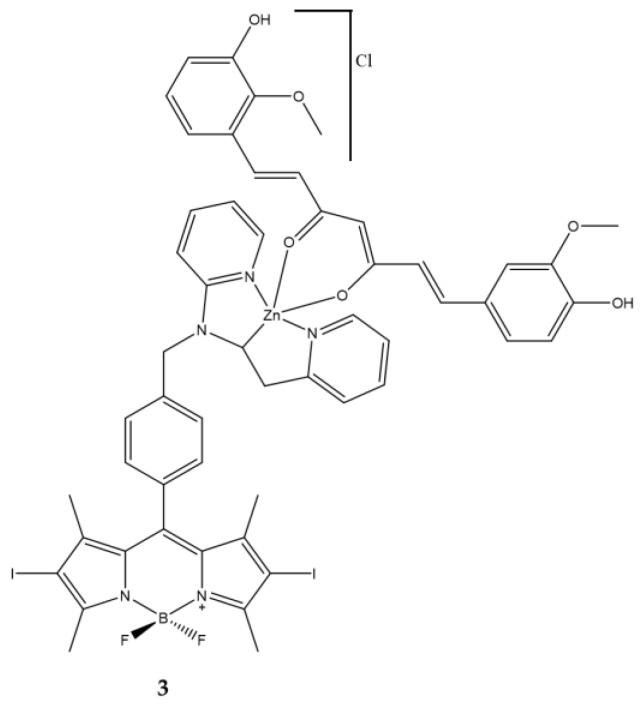
Chemical structures of compound **3** [122,123].

**Figure 15 ijms-23-10198-f015:**
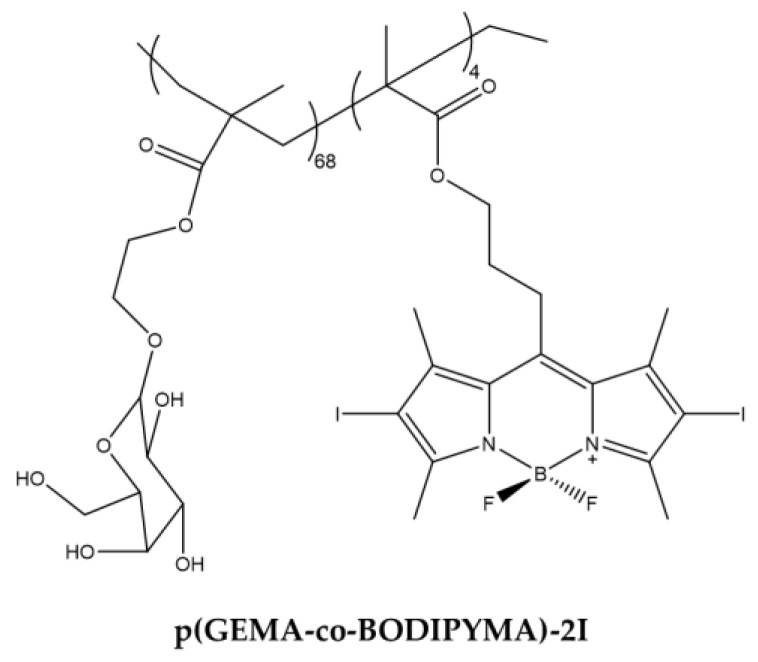
Chemical structures of compound **p(GEMA-co-BODIPYMA)-2I** [111].

**Figure 16 ijms-23-10198-f016:**
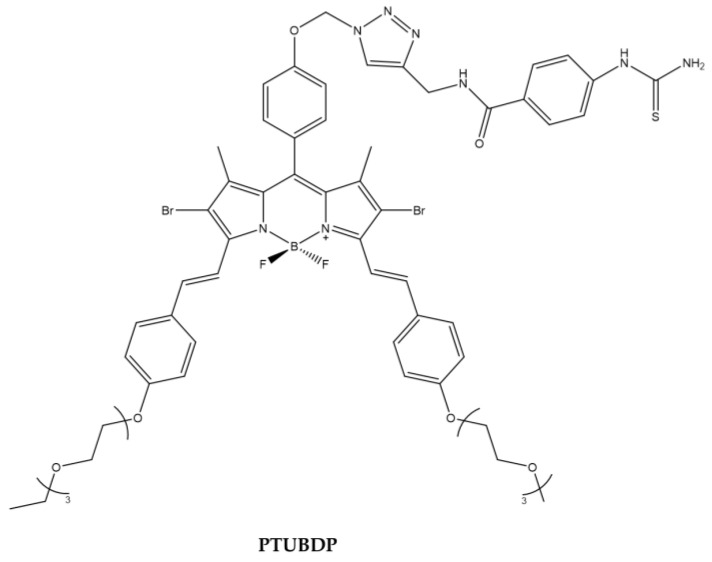
Chemical structures of compound **PTUBDP** [129].

**Figure 17 ijms-23-10198-f017:**
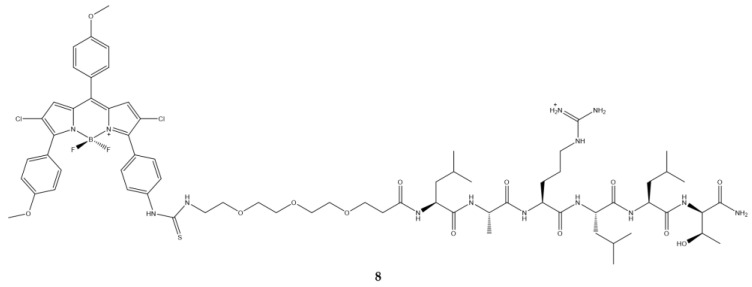
Chemical structures of compound **8** [131].

**Figure 18 ijms-23-10198-f018:**
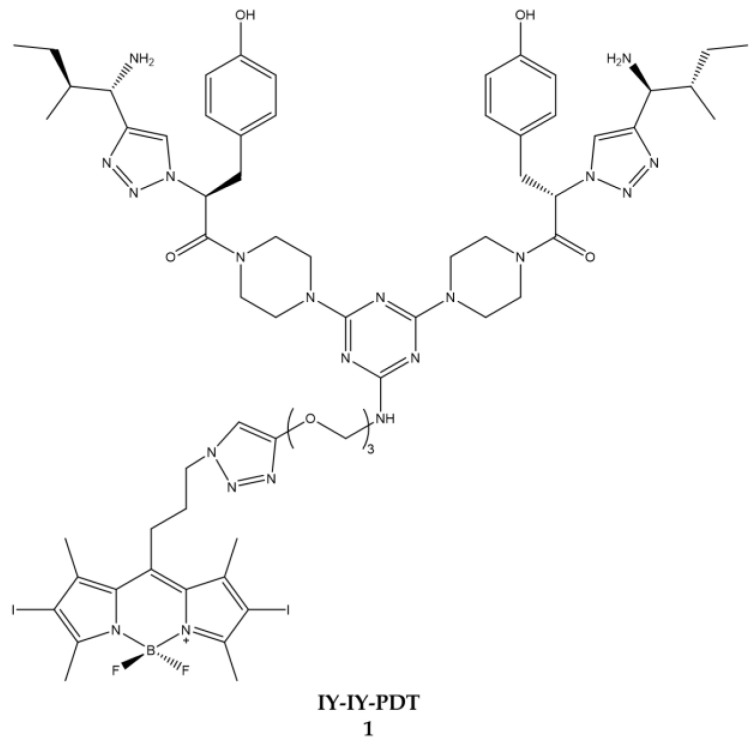
Chemical structures of compound **1** (**IY-IY-PDT**) [135].

**Figure 19 ijms-23-10198-f019:**
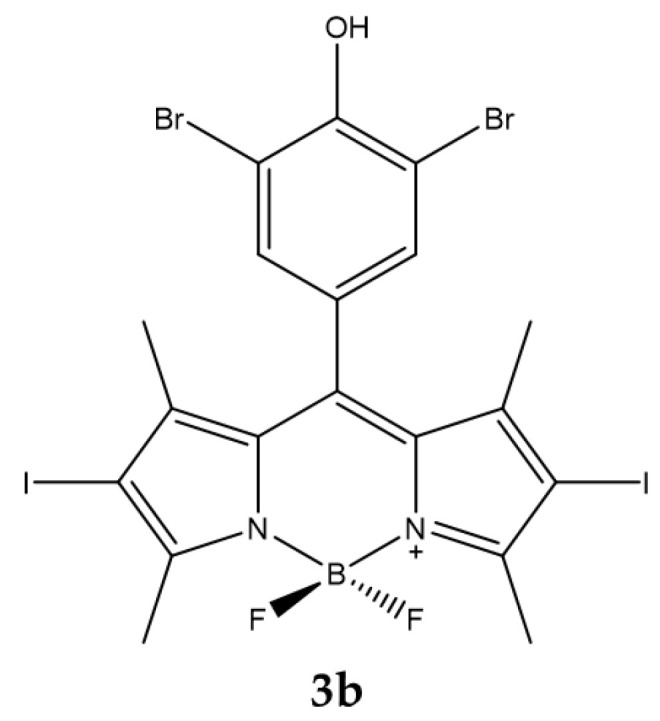
Chemical structures of compound **3b** [138].

**Figure 20 ijms-23-10198-f020:**
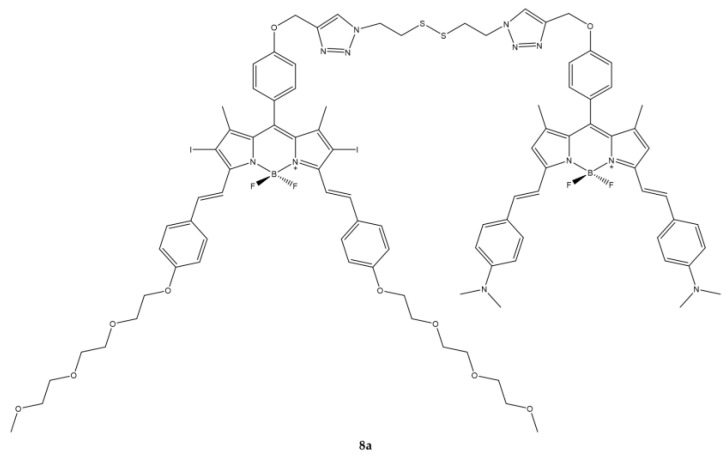
Chemical structures of compound **8a** [139].

**Figure 21 ijms-23-10198-f021:**
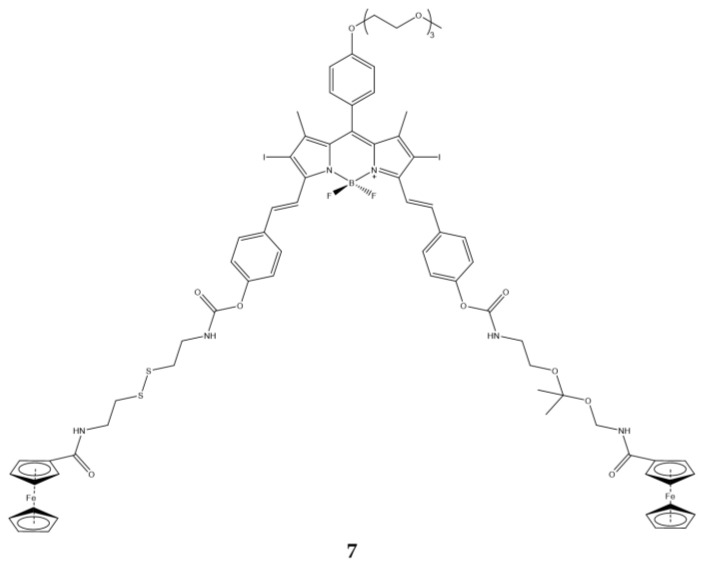
Chemical structures of compound **7** [140].

**Table 1 ijms-23-10198-t001:** Physico-chemical properties and biological data obtained for compounds **2** and **8** [86].

	2	8
λ_abs_ ^a^	534 nm	548 nm
ε	61,500 M^−1^cm^−1^	76,400 M^−1^cm^−1^
Φ_fluo_ ^b^	0.02	0.02
^1^O_2_ QY ^c^	1.04	1.53
IC_50 (SKOV3)_ ^d^	0.65 nM	0.77 nM

^a^ 10 μM solution in Dichloromethane (DCM). ^b^ In DCM with fluorescein (0.1 M NaOH, 0.85) as standard. ^c^ In isopropanol with DPBF (1,3-diphenylisobenzofuran) as indicator and Rose Bengala as standard. ^d^ 2 h of irradiation with a green LED device (fluence rate 25.2 J/cm^2^).

**Table 2 ijms-23-10198-t002:** Physico-chemical properties and biological data obtained for compounds **3** and **4** [87].

	3	4
λ_abs_ ^a^	533 nm	534 nm
ε	66,072 M^−1^cm^−1^	41,500 M^−1^cm^−1^
λ_em_ ^a^	553 nm	554 nm
Φ_fluo_ ^b^	0.01	0.01
^1^O_2_ QY ^c^	1.10	1.11
IC_50 (HCT116)_ ^d^	2.0 nM	11.7 nM
IC_50 (SKOV3)_ ^d^	3.56 nM	15.61 nM
IC_50 (MCF7)_ ^d^	6.88 nM	23.11 nM

^a^ 10 μM solution in DCM. ^b^ In DCM with fluorescein (0.1 M NaOH, 0.85) as standard. ^c^ In isopropanol with DPBF as indicator and Rose Bengala as standard. ^d^ 2 h of irradiation with a green LED device (fluence rate 25.2 J/cm^2^).

**Table 3 ijms-23-10198-t003:** Physico-chemical properties and biological data obtained for compound **3** [89].

	3
λ_abs_ ^a^	528 nm
ε	75,500 M^−1^cm^−1^
λ_em_ ^a^	546 nm
Φ_fluo_ ^b^	0.02
^1^O_2_ QY ^c^	0.93
IC_50 (LLC)_ ^d^	10.0 μM

^a^ In acetonitrile. ^b^ Determined by a standard relative method with Rhodamine 6 G (Φ_fluo_ ≈ 0.94 in ethanol) as a reference. ^c^ In acetonitrile using acridine as standard. ^d^ Irradiation with a light dose of 3.5 mW/cm^2^ from a filtered light source (522 ± 40 nm) for different time.

**Table 4 ijms-23-10198-t004:** Physico-chemical properties and biological data obtained for compounds **Br_2_BOAc** and **I_2_BOAc** [90,91].

	Br_2_BOAc	I_2_BOAc
λ_abs_ ^a^	543 nm	550 nm
ε	81,000 M^−1^cm^−1^	96,000 M^−1^cm^−1^
λ_em_ ^a^	562 nm	572 nm
Φ_fluo_ ^b^	0.14	0.02
^1^O_2_ QY ^c^	0.84	0.98
IC_50 (HeLa)_ ^d^	140.0 nM	180.0 nM

^a^ In acetonitrile. ^b^ Determined by the consumption of dimethylanthracene using Rose Bengal as a standard. ^c^ In acetonitrile using Rose Bengala as standard. ^d^ Irradiation with a 520 nm LED panel for 30 min.

**Table 5 ijms-23-10198-t005:** Physico-chemical properties and biological data obtained for compounds **6** (**PYR**) and **7** (**EPO**) [93].

	6 (PYR)	7 (EPO)
λ_abs_ ^a^	668 nm	667 nm
ε	22,200 M^−1^cm^−1^	74,200 M^−1^cm^−1^
λ_em_ ^a^	697 nm	700 nm
Φ_fluo_ ^b^	0.14	0.10
^1^O_2_ QY ^c^	0.15	0.20
IC_50 (HeLa)_ ^d^	49.0 nM	8.6 nM

^a^ In DMSO. ^b^ In DMSO with Cresyl Violet (Φ_fluo_ ≈ 0.66 in methanol) as a reference. ^c^ In ethanol using Methylene blue as a reference (^1^O_2_ QY ≈ 0.52 in ethanol). ^d^ Irradiation cycles of light (λ = 655 nm, 10 min) and dark (50 min) with a total of 24 h of light exposure.

**Table 6 ijms-23-10198-t006:** Physico-chemical properties and biological data obtained for compounds **4** (**BDPI**) and **6** (**BBDPI**) [102].

	4 (BDPI)	6 (BBDPI)
λ _abs_ ^a^	533 nm	540 nm
^1^O_2_ QY ^b^	0.73	0.68
IC_50 (HeLa)_ ^c^	1.0 μM	2.8 μM

^a^ In DCM. ^b^ Using DPBF as the probe and methylene blue (ΦΔ = 0.57 in DCM) as standard. ^c^ Irradiation with a xenon lamp (40 mW/cm^2^) for 8 min.

**Table 7 ijms-23-10198-t007:** Physico-chemical properties and biological data obtained for compounds **2a**–**d** [103].

	2a	2b	2c	2d
λ_abs_ ^a^	521 nm	507 nm	538 nm	507 nm
λ_em_ ^a^	563 nm	540 nm	568 nm	542 nm
Φ_fluo_ ^b^	0.007	0.002	0.002	0.002

^a^ In DCM. ^b^ In DCM with fluorescein (0.1 M NaOH, 0.90) as standard.

**Table 8 ijms-23-10198-t008:** Physico-chemical properties and biological data obtained for compound **4** [68].

	4
λ_abs_ ^a^	573.8 nm
ε	8,2000 M^−1^cm^−1^
λ_em_ ^a^	590 nm
Φ_fluo_ ^b^	0.6
IC_50 (A549)_ ^c^	2.7 μM

^a^ In ethanol. ^b^ Relative to that of rhodamine 101 (Φ_fluo_ = 1.0 in EtOH). ^c^ Irradiation for 80 min (irradiance: 0.77 mW/cm^2^).

**Table 9 ijms-23-10198-t009:** Physico-chemical properties and biological data obtained for compound **BDILa** [112].

	BDILa
λ_abs_ ^a^	526 nm
ε	41,800 M^−1^cm^−1^
λ_em_ ^a^	542 nm
Φ_fluo_ ^b^	0.02
^1^O_2_ QY ^c^	0.47
IC_50 (HeLa)_ ^d^	0.55 μM
IC_50 (MCF7)_ ^d^	0.61 μM
IC_50 (Huh7)_ ^d^	0.50 μM

^a^ In H_2_O. ^b^ In methanol. ^c^ In ethanol. ^d^ Irradiation for 20 min with a green LED (λ = 530 nm).

**Table 10 ijms-23-10198-t010:** Physico-chemical properties and biological data obtained for compound **BDPI-Lyso** [115].

	BDPI-Lyso
λ_abs_ ^a^	545 nm
ε	41,900 M^−1^cm^−1^
λ_em_ ^a^	572 nm
Φ_fluo_ ^a^	0.05
^1^O_2_ QY ^a^	0.95
^1^O_2_ QY_pH = 5_ ^b^	0.51
^1^O_2_ QY_pH = 7_ ^c^	0.38
IC_50 (Bel-7402)_ ^d^	0.4 μM

^a^ In ethanol. ^b^ In ethanol/PBS = 1:1, pH = 5.10. ^c^ In ethanol/PBS = 1:1, pH = 7.24. ^d^ Irradiation for 30 min with green light (λ = 555 nm, light dose: 4 mW/cm^2^).

**Table 11 ijms-23-10198-t011:** Physico-chemical properties and biological data obtained for compound **MBDP** [119].

	MBDP
λ_abs_ ^a^	660 nm
ε	83,226 M^−1^cm^−1^
λ_em_ ^a^	694 nm
Φ_fluo_ ^b^	0.11
^1^O_2_ QY ^c^	0.64
IC_50 (MCF7)_ ^d^	0.2 μM

^a^ In DCM. ^b^ In DCM with Cresyl Violet (Φfluo ≈ 0.66 in methanol) as a reference. ^c^ Using DPBF as the ^1^O_2_ capture agent and methylene blue as standard (Φ = 0.57 in DCM). ^d^ Irradiation with 660 nm LED red light (20 mW/cm^2^, 48 J/cm^2^).

**Table 12 ijms-23-10198-t012:** Physico-chemical properties and biological data obtained for compound **10** [121].

	10
λ_abs_ ^a^	669 nm
ε	114,815 M^−1^cm^−1^
λ_em_ ^a^	692 nm
Φ_fluo_ ^b^	0.32
^1^O_2_ QY ^c^	0.11
IC_50 (Hela)_ ^d^	0.09 μM
IC_50 (HepG2)_ ^d^	0.16 μM

^a^ In PBS with 0.3% *v*/*v* Tween 80 and 1% *v*/*v* DMF. ^b^ With reference to zinc(II) phthalocyanine (Φ_fluo_ = 0.28 in DMF). ^c^ With reference to zinc(II) phthalocyanine (^1^O_2_ QY ≈ 0.56 in DMF). ^d^ Irradiation with light (λ = 610 nm, 10 min) and dark (50 min) with a total of 24 h of light exposure.

**Table 13 ijms-23-10198-t013:** Physico-chemical properties and biological data obtained for compound **3** [122,123].

	3
λ_abs_ ^a^	543 nm
ε	29,400 M^−1^cm^−1^
λ_em_ ^a^	506 nm
Φ_fluo_ ^b^	0.02
^1^O_2_ QY ^c^	0.73
IC_50 (HeLa)_ ^d^	0.025 μM
IC_50 (MCF7)_ ^d^	0.055 μM
IC_50 (HPL1D)_ ^d^	0.230 μM

^a^ In 1% DMSO-DPBS buffer medium (pH = 7.4). ^b^ With fluoresceinf (0.1 M NaOH, Φ_fluo_ = 0.79) as standard. ^c^ Using Rose Bengal as standard in DMSO (^1^O_2_ QY = 0.76). ^d^ Irradiation with a visible light source for 1 h (fluence rate = 2.4 mW/cm^2^, light dose = 10 J/cm^2^).

**Table 14 ijms-23-10198-t014:** Physico-chemical properties and biological data obtained for compound **p(GEMA-co-BODIPYMA)-2I** [111].

	p(GEMA-co-BODIPYMA)-2I
λ_abs_ ^a^	535 nm
^1^O_2_ QY ^b^	0.79

^a^ In H_2_O. ^b^ Rose Bengal (^1^O_2_ = 0.76) was used as reference.

**Table 15 ijms-23-10198-t015:** Physico-chemical properties and biological data obtained for compound **PTUBDP** [129].

	PTUBDP
λ_abs_ ^a^	667 nm
ε	54,400 M^−1^cm^−1^
λ_em_ ^a^	712 nm
^1^O_2_ QY ^b^	0.093

^a^ In DMSO. ^b^ In DMSO with Rose Bengala as a reference.

**Table 16 ijms-23-10198-t016:** Physico-chemical properties and biological data obtained for compound **8** [131].

	8
λ_abs_ ^a^	588 nm
ε	43,000 M^−1^cm^−1^
λ_em_ ^a^	634 nm
Φ_fluo_ ^b^	0.003
IC_50 (HepG2)_ ^c^	74 μM

^a^ In DMSO. ^b^ Cresyl Violet (Φfluo ≈ 0.50 in ethanol) as a reference. ^c^ Irradiation with halogen lamp for 20 min with a light dose of 1.5 J/cm^2^.

**Table 17 ijms-23-10198-t017:** Physico-chemical properties and biological data obtained for compound **1** (**IY-IY-PDT**) [135].

	IY-IY-PDT
λ_abs_ ^a^	532 nm
λ_em_ ^a^	550 nm
IC_50 (NIH3T3)_ ^b^	0.35 μM

^a^ In DMSO. ^b^ Irradiation with a halogen lamp for 10 min with a light dose of 7.3 J/cm^2.^

**Table 18 ijms-23-10198-t018:** Physico-chemical properties and biological data obtained for compound **3b** [138].

	3b
λ_abs_ ^a^	534 nm
ε	99,000 M^−1^cm^−1^
λ_em_ ^a^	550 nm
Φ_fluo_	0.02
^1^O_2_ QY ^b^	0.59
IC_50 (HeLa)_ ^c^	70 nM
IC_50 (HeLa, pH=5.5)_ ^d^	30 nM
IC_50 (HeLa, pH=7.5)_ ^d^	150 nM

^a^ In acetonitrile. ^b^ With DPBF as indicator and Rose Bengala as standard. ^c^ Irradiation with laser diode (λ = 532 nm) for 30 min. ^d^ Irradiation with laser diode (λ = 532 nm) for 5 min with an irradiance of 1 mW/cm^2^.

**Table 19 ijms-23-10198-t019:** Physico-chemical properties and biological data obtained for compound **8a** [139].

	8a
λ_abs_ ^a^	664 nm
λ_em_ ^a^	768 nm
Φ_fluo_ ^b^	0.052
^1^O_2_ QY ^c^	0.018
IC_50 (HeLa)_ ^d^	0.67 μM
IC_50 (A549)_ ^d^	0.44 μM
IC_50 (H22)_ ^d^	0.48 μM

^a^ In DMF. ^b^ Relative to unsubstituted zinc(II) phthalocyanine in DMF as the reference (Φfluo ≈ 0.28). ^c^ Relative to unsubstituted zinc(II) phthalocyanine (^1^O_2_ QY ≈ 0.56). ^d^ Irradiation with a 670 nm LED lamp for 2 min with an irradiance of 20 mW/cm^2^.

**Table 20 ijms-23-10198-t020:** Physico-chemical properties and biological data obtained for compound **7** [140].

	7
λ_abs_ ^a^	376, 441, 662 nm
λ_em_ ^a^	686 nm
Φ_fluo_ ^a^	0.03
IC_50 (MCF7, no DTT_ ^b^	146.0 nM
IC_50 ((MCF7, 2 μM DTT)_ ^b^	140.0 nM
IC_50 (MCF7, 4 mM DTT)_ ^b^	81 nM

^a^ In PBS with 0.25% Cremophor EL. ^b^ Irradiation with a halogen lamp for 20 min with a total fluence of 48 J/cm^2^.

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
