# Peer review of "BODIPYs in PDT: A Journey through the Most Interesting Molecules Produced in the Last 10 Years"

_ijms, 2022, doi:10.3390/ijms231710198_

Round 1

Reviewer 1 Report (Previous Reviewer 2)

Dear

Great effort of correction by several corrections yet

Best regards

Reviewer 2 Report (Previous Reviewer 1)

The manuscript submitted by Enrico Caruso and co-workers has been corrected following the advice of the reviewers and can now be published in IJMS.

This manuscript is a resubmission of an earlier submission. The following is a list of the peer review reports and author responses from that submission.

Round 1

Reviewer 1 Report

The review sumitted by E. Caruso and co-workers is interesting and can be published in IJMS however with major modification listed below:

Some reviews already cover the field and have to be cited: https://doi.org/10.1021/acs.chemrev.1c00381

For example figure 6 has already been described. The authros should therefore  take previous reviews into account and revise their manuscript accordingly. 

Concerning azabodipy, review:

DOIhttps://doi.org/10.1039/D0CS00234H

can be cited,

Author Response

We thank the reviewer for the suggestions given and based on his indications we have integrated the bibliography by inserting several new bibliographic references relating to some reviews published in these two years dealing with BODIPY.
As for the AzaBodipy, the authors chose not to introduce them in the review, considering them a subclass of the BODIPY, consequently, while considering the suggestion very valid, the authors preferred not to include the topic.

Reviewer 2 Report

Dear

Malacarne et al give a overview of BODIPY derivatives and their effects in vitro and in vivo. However, the paper presented major flaws which could be easily corrected and will improve the text.

Major points

-  Absence of tables which summarize the characteristics of the derivatives and their impact on cells ( in vitro or in vivo). The tables will help the lector and give  precision which are missing in the text. For an example, whether a BODIPY derivative gives a higher singlet oxygen yield, the calculated value could be stated in the table. Such absence of precision exists in the text and might be reconsidered (origin of the cell line used for another example).

- The authors often detailed the chronology of each project (publication after another publication of the same group), starting from the physical characteristics, then the result obtained with cell and or not in vivo.  The authors are encouraged to summarize the corresponding text (examples are given in the pdf)

- check also the legend of the figure: the synthesized compound belongs to a serie presented n each publication. Clearly, the authors must add the corresponding reference in each legend

- The text must be edited again

Author Response

Major points

- Absence of tables which summarize the characteristics of the derivatives and their impact on cells (in vitro or in vivo). The tables will help the lector and give precision which are missing in the text. For an example, whether a BODIPY derivative gives a higher singlet oxygen yield, the calculated value could be stated in the table. Such absence of precision exists in the text and might be reconsidered (origin of the cell line used for another example).

The authors at the suggestion of the reviewer entered the tables for each molecule covered in the review. The tables summarize the chemical-physical and biological data present in each reported article.

- The authors often detailed the chronology of each project (publication after another publication of the same group), starting from the physical characteristics, then the result obtained with cell and or not in vivo. The authors are encouraged to summarize the corresponding text (examples are given in the pdf)

At the suggestion of the reviewer and following all the indications given in the attached pdf, the authors have summarized the text in many parts (also in consideration of the added tables).

- check also the legend of the figure: the synthesized compound belongs to a serie presented n each publication. Clearly, the authors must add the corresponding reference in each legend

Bibliographic notes relating to the article cited have been inserted in all the captions of the figures, the same type of editing has also been done in the added tables.

- The text must be edited again

the whole work was corrected from the point of view of editing and English. The various errors present have been corrected.

All the indications given in the attached pdf file have been corrected as suggested by the reviewer. An initial explanatory figure of the photodynamic process has been added, the photodynamic process has been better clarified as required, the texts have been summarized as indicated, the unclear sentences have been fixed, the uptake and targeting paragraph has been reorganized into subparagraphs, differentiating better uptake and targeting. The only change not made was the request for a paragraph of the conclusions. The journal does not consider it necessary especially in the case of reviews and the authors did not consider it necessary to add it at the end of the review.

Round 2

Reviewer 1 Report

The manuscript has been revised following the advice of the reviewers and can be published as is.

Author Response

Thank you

Reviewer 2 Report

Dear

PLesae because of the many changes that have been proposed, the authors should provide  the new version without corrections embedded in the text.

Best regards

Author Response

As you can see in the attached pdf file we highlighted all the requested revisions

Best regards

Round 3

Reviewer 2 Report

Dear,

Cf the pdf version of the manuscript
